# Levels, Distributions, and Potential Risks of Hexachlorobutadiene from Two Tetrachloroethylene Factories in China

**DOI:** 10.3390/ijerph20065107

**Published:** 2023-03-14

**Authors:** Chengyou Liu, Jing Guo, Meng Liu, Jinlin Liu, Lifei Zhang

**Affiliations:** 1State Environmental Protection Key Laboratory of Dioxin Pollution Control, National Research Center for Environmental Analysis and Measurement, Beijing 100029, China; 2Environmental Development Center, Ministry of Ecology and Environment, Beijing 100029, China

**Keywords:** hexachlorobutadiene, tetrachloroethylene, level, risk, management

## Abstract

A systematic investigation was conducted on the emission of hexachlorobutadiene (HCBD) from two tetrachloroethylene factories that used the acetylene method (F1) and the tetrachloride transformation method (F2). The levels of HCBD in the air for F1 were found to be in the range of 1.46–1170 µg/m^3^, whereas F2 had levels in the range of 1.96–5530 µg/m^3^. Similarly, the levels of HCBD in the soil for F1 were found to be in the range from 42.2 to 140 µg/kg, whereas F2 had levels in the range from 4.13 to 2180 µg/kg. Samples obtained from the air, soil, and sludge in the reaction area of the tetrachloroethylene factories in China showed high levels of HCBD. The F1 method unintentionally produced more HCBD than the F2 method during tetrachloroethylene production, leading to greater harm. The results of the risk assessment suggested the presence of harmful health effects on workers in the workplace. The investigation findings highlight the need for improved management systems to ensure the safe production of tetrachloroethylene.

## 1. Introduction

Hexachlorobutadiene (HCBD) is a halogenated aliphatic compound, mainly created as a by-product in the manufacture of chlorinated aliphatic compounds. It was first synthesized in 1877 by the chlorination of hexyl iodide [1]. HCBD has been detected in various abiotic and biotic media such as water [2], air [3], sediment [4], soil [5,6,7], and organisms [8]. Low HCBD concentrations have also been measured in Arctic terrestrial and marine biota [9]. Thus, HCBD has been classed as a priority pollutant in many countries. In 2015, HCBD was listed in Annex A as a persistent organic pollutant (POP) by the Stockholm Convention for its persistence, toxicity, bioaccumulation, and long-range transport abilities [10].

It seems that HCBD is no longer intentionally produced and used in the United Nations Economic Commission for Europe, including in the United States and Canada [10]. HCBD was also not intentionally produced in large commercial quantities in China [5]. However, it can be unintentionally produced during magnesium production [11] and some chemical production processes, such as the production of carbon tetrachloride and tetrachloroethylene [12], and the production of chlorinated methane [13].

Tetrachloroethylene, also called perchloroethylene, is most commonly known for its wide use in the dry-cleaning industry and as a chemical intermediate in the production of several fluorinated compounds. It is a basic raw material in the manufacture of hydrofluorocarbons 134a and 125, which are alternatives to chlorofluorocarbon and hydrochlorofluorocarbon refrigerants [14]. In China, tetrachloroethylene is mainly manufactured by two different synthetic routes: the acetylene method (reaction 1–5) and the carbon tetrachloride conversion method (reaction 6) as shown below. Due to the restrictions of the Montreal Protocol and the high incineration cost of carbon tetrachloride, the latter method is usually used to produce chlorinated methanes.
CaC_2_ + 2H_2_O → Ca(OH)_2_ + C_2_H_2_(1)
C_2_H_2_ + 2Cl_2_ → CHCl_2_CHCl_2_(2)
CHCl_2_CHCl_2_ → CHClCCl_2_ + HCl(3)
CHClCCl_2_ + Cl_2_ → CHCl_2_CCl_3_(4)
CHCl_2_CCl_3_ → CCl_2_CCl_2_ + HCl(5)
2CCl_4_ + 2H_2_ → CCl_2_CCl_2_ + 4HCl(6)

The latter method, regardless of kind of method, chlorinated alkanes, chlorinated alkenes, and chlorinated aromatic compounds would be produced more or less during these chemical reaction processes. Tetrachloroethylene and trichloroethyleneproduced in industrial processes are separated by their boiling points. In this case, other high boiling point substances such as hexachlorobenzene and HCBD may be emitted to the surrounding environment, especially into the air, intermediate products, and some residues. Unfortunately, few studies have investigated the emissions of HCBD from tetrachloroethylene factories [12]. In this study, a total of 28 samples including 10 ambient air samples, 6 soil samples, one sludge sample, and some residue and product samples, from two tetrachloroethylene factories in China, were analyzed to investigate the level and distribution of HCBD contamination. The risk of occupational exposure to HCBD in the workplace was assessed during this chemical production process. Consequently, recommendations were made for the safety risk management of industrial tetrachloroethylene production. The results of this research will facilitate the implementation of China’s national implementation plan for the Stockholm Convention.

## 2. Materials and Methods

### 2.1. Chemicals and Reagents

The standard solutions of HCBD and 4-bromofluorobenzene were purchased from Accustandard Inc. (New Haven, CT, USA). Carbon disulfide (ultra low in benzene, for trace analysis, ≥99.9%) and methanol were purchased from CNW Technologies GmbH (Düsseldorf, Germany). Water was purified with a Milli-Q Integral water purification system from Merck Millipore (Darmstadt, Germany).

### 2.2. Factories and Samples

The two tetrachloroethylene factories are located in Inner Mongolia Autonomous Region (F1) and Shandong Province (F2) respectively (Figure 1). F1 uses the acetylene method (see reaction 1 to 5) for tetrachloroethylene production. F2 is a methanol-based chlorinated methane production factory. The carbon tetrachloride transformation method (see reaction 6) is used for tetrachloroethylene production in F2.

As shown in Figure 1, ambient air samples were collected based on different functional areas from 30 August to 4 September. Briefly, sampling was performed by using a TH-110F low-volume air sampler from Wuhan Tianhong Instruments Co., Ltd. (Wuhan, China) at a flow rate of 1.0 L/min with 60 min of sampling intervals. Field blanks were obtained by placing the activated charcoal tube into the sampler head without sucking. Soil samples were collected in factories from different positions (Figure 1). Only one sludge sample generated from rain wash was obtained from F2. The commercial products of trichloroethylene and tetrachloroethylene and intermediate products such as carbon tetrachloride, tetrachloroethane, and pentachloroethane, and some residues such as high boiling liquid or crystalline precipitate were also collected for HCBD analysis.

### 2.3. Ambient Air Sample Analysis

Ambient air samples were determined using the activated charcoal adsorption and carbon disulfide desorption method. The ambient air samples were transported to the laboratory immediately after sampling. Then, the activated charcoal was desorbed into a GC auto-injector vial with 1.0 mL carbon disulfide for 30 min. The desorption solution was used for HCBD analysis. The determination of HCBD was performed on a Shimadzu GCMS-QP2010plus (Kyoto, Japan) equipped with a fused silica capillary DB-5MS column (30 m × 0.25 mm i.d., film thickness 0.25 µm) using electron ionization with selective ion monitoring mode. High purity (99.999%) helium was used as the carrier gas at 1.0 mL/min. The injector, transfer line, and ion source temperatures were 270, 280, and 230 °C, respectively. The oven temperature program was as follows: initial temperature 50 °C, held for 2 min; increased to 120 °C at 20 °C/min, held for 3 min; increased to 160 °C at 10 °C/min; and increased to 300 °C at 30 °C/min. Two microliters of each sample were injected with 10:1 in split mode. The ions of HCBD selected for quantification and identification were 225, 227, 258, and 260. Qualitative identification was based on retention time analysis. Mass spectral verification on real samples was carried out by comparison of relative abundance values of the quantification and qualification ions to the same values obtained from the standards. Total sample volumes for the air samples were about 60 L. The method detection limit for HCBD in air was 0.6 µg/m^3^. No HCBD was detected in the blank samples.

### 2.4. Soil and Sludge Sample Analysis

Soil and sludge samples were determined using the purge and trap method. Briefly, 5 g of sample was fortified with internal standard compound (4-bromofluorobenzene), mixed with 5.0 mL of water, and purged at room temperature on an Eclipse 4660 system from OI analytical (USA). Purged compounds were trapped on a No. 10 trap (Tenax/silica gel/carbon molecular sieve sorbent) and then thermally desorbed and separated on a DB-624 column (60 m × 0.32 mm i.d., film thickness 1.8 µm) with a Shimadzu GCMS-QP2010 (Kyoto, Japan) using electron ionization with a selective ion monitoring mode. High purity (99.999%) helium was used as the carrier gas at 1.58 mL/min. The injector, transfer line, and ion source temperatures were 220, 240, and 200 °C, respectively. The oven temperature program was as follows: initial temperature 50 °C, held for 5 min; increased to 120 °C at 7 °C/min; increased to 240 °C at 12 °C/min, held for 5 min. The method recovery for HCBD was 112%. The method detection limit for HCBD in soil and sludge was 0.6 µg/kg.

### 2.5. Residue and Product Sample Analysis

Residue and product samples were determined using the direct injection method. The residues include tetrachloroethane, pentachloroethane, carbon tetrachloride, trichloroethylene low-boiling substance, tetrachloroethylene heavy fraction, trichloroethylene desorption, and tetrachloroethylene desorption. Among all of the samples, only tetrachloroethylene heavy fraction is crystal. Other residue and product samples are liquid. A certain quality of the sample was dissolved with carbon disulfide and analyzed with GCMS. The rest of the process is as described for ambient air samples. The method detection limit for HCBD in residue and product was 1.0 µg/kg.

## 3. Results and Discussion

### 3.1. Levels and Distributions of HCBD in Air, Soil, and Sludge Samples

Concentrations of HCBD in air, soil, and sludge samples are given in Table 1. HCBD was detected in all of the samples, indicating that it is widely distributed in the environment of tetrachloroethylene factories. Air is the main environmental compartment in which HCBD is found according to several sources [9]. About 95% of the emitted HCBD remains in the atmosphere and could be transported for redistribution [15,16]. The HCBD levels in the air were in the range of 1.46–1170 µg/m^3^ (median 179 µg/m^3^) for F1 and 1.96–5530 µg/m^3^ (median 13.2 µg/m^3^) for F2. Air levels of HCBD in this study were much higher than those at Alert (a high Arctic station in Nunavut, Canada) [9], and comparable to those found in the workplace in Baoshan District, Shanghai (216–896 µg/m^3^) [17] and those detected in the area near Fong Shan Stream, a heavily polluted river in Taiwan Province (226 µg/m^3^) [3]. The results indicated that high levels of HCBD in the air can be found in the tetrachloroethylene production factories.

The distributions of HCBD in the air in the two factories were extremely non-uniform. Interestingly, the lowest and the highest HCBD values in air were detected from the upper wind direction (air 1 for F1, air 3 for F2) and the reaction area (air 3 and air 4 for F1, air 1 and air 4 for F2) for both of the two factories. Air samples collected from the storage area had similar HCBD results, with 8.59 (air 2) and 8.20 (air 2) µg/m^3^ for F1 and F2, respectively. The investigation results indicated that the reaction area was the main emission source of HCBD in the air in tetrachloroethylene production factory. In addition, higher atmospheric HCBD levels were mainly detected in F1 indicating that the acetylene method may unintentionally produce more HCBD than the carbon tetrachloride transformation method during tetrachloroethylene production. As a matter of fact, the acetylene method (reaction 1–5) was started with acetylene, a two-carbon atom compound, which was more prone to produce even numbers of halogenated aliphatic compounds, such as HCBD.

The HCBD levels in the surface soil were in the range of 42.2–140 µg/kg for F1 and 4.13–2180 µg/kg for F2. The results in the soil samples were higher than those soil samples (0.04–27.9 µg/kg) from a chlor-alkali chemical plant in Jiangsu Province [5] and soil samples (<0.02–5.6 µg/kg) from a former pesticide-producing area in Chongqing City [7] in China. The results were much higher than those soil samples (<0.02–3.1 µg/kg) from an agricultural area in Jiangsu Province [6] in China, while being lower than those in soil samples (6400–980,000 µg/kg) from plants producing perchoroethylene and trichloroethylene in United States [18]. This could suggest a point source pollution of HCBD in the environment in China. As shown in Figure 1 and Table 1, the highest concentration of HCBD (2180 µg/kg) was found in the surrounding soil from the reaction area, indicating that the reaction area was the main emission source of HCBD in soil in tetrachloroethylene production factories. The surface soil of the reaction area in a chemical production factory was usually pretreated with cement solidification. In this case, pollutants may be transferred to the surrounding soil due to artificial or physical factors, such as manual cleaning and rain wash. An HCBD level of 72,700 µg/kg found in the sludge sample from F2 confirmed the HCBD emission from the reaction area. Intensive studies were needed to clarify the HCBD pollution status around the tetrachloroethylene production factories.

### 3.2. Levels of HCBD in Residues and Products

Tetrachloroethylene is mainly manufactured using two different synthetic methods, which leads to different intermediate products and residues. As shown in Table 1, two intermediate products (tetrachloroethane and pentachloroethane) and three residue samples (trichloroethylene desorption column liquid, trichloroethylene low boiling liquid, and tetrachloroethylene desorption column liquid) were collected from F1 (acetylene method, reaction 1–5) for HCBD analysis. HCBD was detected in all five samples, with levels ranging from 0.062 to 44,900 mg/kg. For F1, the highest HCBD level was found in the liquid of the trichloroethylene desorption column, in which HCBD content reached 4.5%. For F2, the level of HCBD in the intermediate product, carbon tetrachloride, was 12.9 mg/kg, while HCBD levels in the tetrachloroethylene heavy fraction crystal and liquid reached 78,600 and 84,700 mg/kg, respectively. The average content of HCBD in the tetrachloroethylene heavy fraction was 8.2%. Tetrachloroethylene production reported that the high-boiling residues from the chlorinolysis of propylene were separated into two categories, namely crude hexachlorobenzene and the mother liquor [12]. HCBD was the major component in the mother liquor. Therefore, the content of HCBD in the mother liquor was relatively high and reached up to 31.3%, which was much higher than the results from our study. The HCBD contents in residues were similar to those in the raw product (5%) from low pressure chlorolysis for the manufacturing of perchloroethylene and carbon tetrachloride, but higher than other processes reported by the POPs Review Committee (0.2–1.11%) [19]. Further research is needed to determine the content of HCBD in the residues or raw products from these processes for future evaluation.

The Chinese Ministry of Industry and Information Technology published the standards on industrial tetrachloroethylene (HG/T 3262-2014) and trichloroethylene (HG/T 2542-2014) in 2002 and 1993, respectively. These two standards specified several items including the purity according to their usage. Requirements on purity for trichloroethylene and tetrachloroethylene were revised and promoted in 2014. According to new standards, the purities were ≥99.6% (type II) and ≥99.9% (type I) for commercial tetrachloroethylene, and ≥99.0%—≥99.3% (type II) and ≥99.5%—≥99.9% (type I) for commercial trichloroethylene. It should be noted that levels of HCBD in commercial products, such as trichloroethylene from F1 and tetrachloroethylene from both factories were below the detection limit. This may be due to the stricter chemical product standards implemented by the Chinese government since 2000 as part of the national implementation plan for the Stockholm Convention in China.

### 3.3. Human Health Risk Assessment

Currently, there are no limit values for HCBD in the Chinese soil environmental quality standards or ambient air quality standards. The HCBD concentration limit was 1.0 mg/kg for uncontaminated soils according to the Chinese standard for soil quality assessment for exhibition sites (HJ 350-2007). Most of the soil samples from the tetrachloroethylene factories were below this limit value, indicating the soils are safe, except for one soil sample from the reaction area of F2. The Standards on the Occupational Exposure Limits for Hazardous Agents in the Workplace (GBZ 2.1-2007) were published by the National Health and Family Planning Commission of China in 2007. The time-weighted average permissible concentration for HCBD is 200 µg/m^3^. Air samples collected near the reaction areas of the two factories exceed the permissible value by 1.5–28 times, indicating that (a) production device volatilization or leakage may happen in factories, which may lead to (b) harmful health effects on workers in the workplace. Therefore, the human health risk of HCBD to the sampling areas was evaluated using the Hazard Quotient (HQ) approach from the POPs Toolkit (see Appendix A) [20]. According to the hazard quotient risk calculation tool, the calculation of HQ includes accidental ingestion of soil, inhalation of contaminated particles, and dermal contact with contaminated soil. Parameters for HQ calculation were quoted from the US EPA [21,22], Health Canada (2004), and the local government’s environmental quality bulletin (2014) (Appendix A). The doses from soil ingestion, particle inhalation, and dermal contact were 1.4 × 10^−8^, 6.2 × 10^−10^, and 9.2 × 10^−8^ mg/kg-day for F1, and 2.1 × 10^−7^, 2.0 × 10^−8^, and 1.4 × 10^−6^ mg/kg-day for F2. The calculated HQ for F1 and F2 were 5.28 × 10^−4^ and 8.28 × 10^−3^ (both less than 0.2), implying very low adverse effects of HCBD in the soil for factory workers. It should be noted that the dose from dermal contact accounts for 86% of the total value for the two factories. Lifetime cancer risks (LCR) from inhaling airborne HCBD were estimated based on equations in our previous study [23]. The intake factor for a worker was calculated to be 0.01 m^3^/kg-day. The cancer slope factor for long-term inhalation of HCBD (0.04 kg-day/mg) and the age-dependent adjustment factor (1, unitless) recommended by the US Environmental Protection Agency (USEPA, 2007) were used. The LCRs from inhaling HCBD were estimated for each air sample for the two factories. As shown in Table 1, most of the LCR values were higher than 1.0 × 10^−6^, suggesting that cancer risk from inhaling airborne HCBD potentially exists in the tetrachloroethylene factories in China. However, it should be noted that these LCR values were based on the most severe conditions. There are some limitations on the LCR values. For instance, these samples were collected in August and September, where the high temperature made HCBD more volatile; the concentration values would be lower in other seasons. Furthermore, we did not distinguish among differing cases, the exposure pattern may vary greatly for workers who worked in the factories or lived near the factories, and it was also depended on the specific meteorological conditions. Nevertheless, the human health risk should be brought to the forefront in the tetrachloroethylene factories in China and around the world.

### 3.4. Management Needs

The two main routes for tetrachloroethylene production in China are the acetylene and carbon tetrachloride transformation methods. It was estimated that 0.1 million tons of tetrachloroethylene was produced in 2015, according to the information from China Chlor-Alkali Industry Association. Two-thirds of the tetrachloroethylene was produced by carbon tetrachloride transformation. Our previous study suggested that more research and improved management systems were needed to ensure that the production of chlorinated methanes can be achieved safely [13]. The production of tetrachloroethylene industry should also be the case for China. However, there is a lack of emission or discharge standards for HCBD from chemical facilities. The revised emission standard for pollutants in the chlor-alkali industry (GB 15581-1995), together with the cleaner production standards for the chlor-alkali industry only covers caustic soda (HJ 475-2009) and polyvinyl chloride (HJ 476-2009). Therefore, major POPs including HCBD unintentionally produced in the production of the tetrachloroethylene industry should be incorporated into the relevant standards for strict management.

Secondly, the main residues were tetrachloroethylene desorption column liquid and tetrachloroethylene heavy fraction for the acetylene method and carbon tetrachloride transformation method, respectively. These residues should be treated by incineration according to relevant regulations. Hence, intensive supervision on the residues’ storage, transportation, and incineration should be precisely managed.

Thirdly, high HCBD levels that were found in the reaction areas imply that measures should be taken during tetrachloroethylene production to avoid pollutants leakage or emission. HCBD and other volatile organic compounds should be fit into the leak detection and repair management system. Finally, the lifetime cancer risk implied that inhaling airborne HCBD was a very important source for workers’ health in the factories. It was important to enhance self-protection awareness, and improve occupational safety and protective measures to reduce the level of occupational hazards in the workplace.

## 4. Conclusions

This study provides the first systematic investigation of hexachlorobutadiene (HCBD) emission from tetrachloroethylene production in China. HCBD was found to be widely distributed and highly variable in the environment of the two tetrachloroethylene factories studied. Elevated levels of HCBD in air, soil, and sludge samples were predominantly detected in the reaction area. The highest HCBD concentrations were observed in the trichloroethylene desorption column liquid for F1 and the tetrachloroethylene heavy fraction for F2. Notably, the levels of HCBD in commercial trichloroethylene and tetrachloroethylene products were below the detection limit. However, the F1 method was observed to inadvertently produce more HCBD than the F2 method during tetrachloroethylene production, resulting in greater environmental and ecological harm. The risk assessment indicated potential harmful health effects on workers with varying degrees in the workplace. The study recommends improved management and safety protocols for enterprises to support China’s national implementation plan for the Stockholm Convention. These findings contribute to a better understanding of the environmental distribution and fate of HCBD in contaminated sites associated with chlorinated methanes and/or tetrachloroethylene production.

## Figures and Tables

**Figure 1 ijerph-20-05107-f001:**
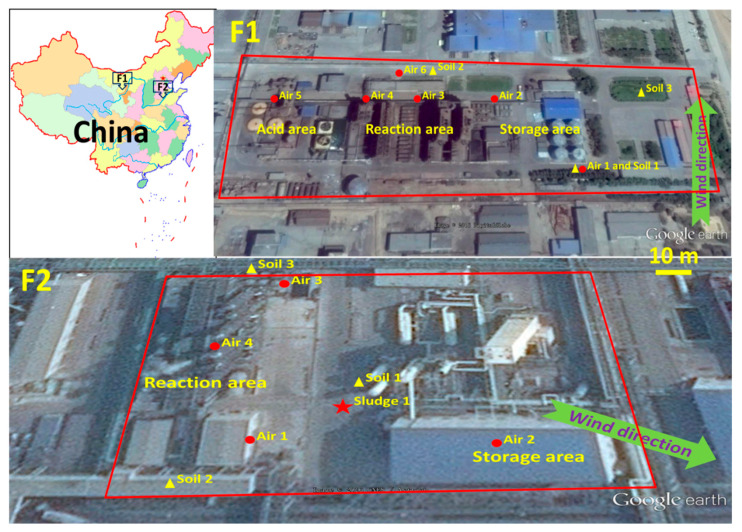
Map showing the air, soil, and sludge sampling sites in two tetrachloroethylene factories in China.

**Table 1 ijerph-20-05107-t001:** Concentrations of HCBD in air, soil, sludge, residue, and product samples from F1 and F2.

Factory	Air			Soil and Sludge			Residue and Product	
	No.	Concentration (µg/m^3^)	LCR (×10^−6^)	No.	Concentration (µg/kg)	HQ (×10^−6^)	Sample Name	Concentration (mg/kg)
**F1**	air 1	1.46	0.584	soil 1	42.2	159	tetrachloroethane	2.72 × 10^3^
	air 2	8.59	3.44	soil 2	140	528	TCE desorption column liquid	4.49 × 10^4^
	air 3	491	196	soil 3	55.3	208	TCE low boiling liquid	0.062
	air 4	1.17 × 10^3^	468	—	—	—	pentachloroethane	3.92
	air 5	53.3	21.3				PCE desorption column liquid	3.80 × 10^3^
	air 6	305	122				commercial TCE and PCE	ND
**F2**	air 1	5.53 × 10^3^	2210	soil 1	2.18 × 10^3^	8280	carbon tetrachloride	12.9
	air 2	8.20	3.28	soil 2	13.3	50.0	PCE heavy fraction crystal	7.86 × 10^4^
	air 3	1.96	0.784	soil 3	4.13	15.0	PCE heavy fraction liquid	8.47 × 10^4^
	air 4	18.1	7.24	sludge 1	7.27 × 10^4^	—	commercial PCE	ND

F1: Factory 1; F2: Factory 2; HCBD: hexachlorobutadiene; LCR: lifetime cancer risk; HQ: hazard quotient; TCE: trichloroethylene; PCE: tetrachloroethylene; ND: not detected.

## Data Availability

Not applicable.

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
