# Peer review of "Levels, Distributions, and Potential Risks of Hexachlorobutadiene from Two Tetrachloroethylene Factories in China"

_ijerph, 2023, doi:10.3390/ijerph20065107_

Round 1
Reviewer 1 Report
1. Section 2.5. What is the residue? Describe more specifically. Is it a liquid, solid or semisolid?
2. Table 1. ND should mean “not detected”, instead of “not detectable”.
3. Section 3.4. It is not necessary to repeat the two methods for tetrachloroethylene production since the production routes are mentioned in the Introduction section. The second paragraph of this section (L. 262 – 267) should be deleted since it is not involved in the HCBD management.
4. Table 1 should be modified to fit the scale of the manuscript. The table legend should be aligned with the header of the table. “Conc” should be replaced by fully spelling “concentration” for a formal publication.
5. The authors mentioned that the HCBD emissions from TCE factories had been investigated in few studies (L. 57-58). The concentrations of HCBD found in this study could compare to those of previous studies to find whether the findings of this study have significant variation.
Reviewer 2 Report
This paper is not suitable for publication due to several concerns.
1. The English is difficult for reading.
2. Just a monitoring of several samples, maybe be better for a local journal.
3. The risk assessment is hard for reading. Why not provide some information in supporting materials? Some parameters should obtained from the real world, not from references. How about the risk when you both consider contact and inhalation?
Reviewer 3 Report
This work studied the air, soil, sludge, products and residue samples from two tetrachloroethylene factories which use two major production methods. The HCBD occurrence and concentrations were analyzed. Human health risk and management suggestions were also proposed. This is the first systematic report on HCBD emission from the tetrachloroethylene production in China. This topic is within the scope of this journal. However, there are some questions and deficiencies in the article that need to be consulted with the author.
1. What steps should be taken to ensure safer production of tetrachloroethylene, and what suggestions are made to reduce the unconscious production of hexachlorobutadiene?
2. What are the potential factors that may contribute to variations in HCBD emissions across different production methods and facilities?
3. Line 57-58, the manuscript have mentioned that emissions of HCBD have been investigated in a few studies, but only one reference was presented.
4. Line 241-245, what is the basis of these assumed values for calculation?
5. The language of part of the manuscript need further polish. There only give a few examples.
1) Line 137-141, the detected concentration of HCBD in the studied area is much higher than that in Taiwan, especially in F2. Thus, it is inappropriate to say "and to those detected in areas near Fong Shan Stream, a heavily polluted river in Taiwan (226 μg/m3 )"
2) Line 175, "which lead to " could be changed to "which leads to".
3) Line 224, the sentence "exceed the permissible value with 1.5-28 times" can be changed to "exceed the permissible value by a factor of 1.5-28"
Round 2
Reviewer 2 Report
I still do not agree to accept this paper since the authors do not respect my comments, especially my comment 3.
In their paper, they stated that "
Our calculations assumed that the exposure time, exposure frequency, exposure duration, and percentage of inhaled chemical absorbed were 8 hours/day, 200 days/year, 20 years, and 100%, respectively." If the risk assessment can be assumed as the authors like, I do not think this is a scientific paper. They even did not do a survey. The people who worked in the factories or lived near the factories have different exposure pattern. What's more, the seasonal difference is not considered, and they did not state this as a limitation.
Only when the authors address these issues, this paper can be accepted.
